# Gastro-Intestinal Microbiota in Equines and Its Role in Health and Disease: The Black Box Opens

**DOI:** 10.3390/microorganisms10122517

**Published:** 2022-12-19

**Authors:** Frédérique Chaucheyras-Durand, Audrey Sacy, Kip Karges, Emmanuelle Apper

**Affiliations:** 1Lallemand SAS, 31702 Blagnac, France; 2UMR MEDIS, INRAE, Université Clermont-Auvergne, 63122 Saint-Genès Champanelle, France; 3Lallemand Specialities Inc., Milwaukee, WI 53218, USA

**Keywords:** equine, gastrointestinal tract, microbiota, dysbiosis, metabolism, Proteobacteria, fibrolytic bacteria

## Abstract

Horses are large non-ruminant herbivores and rely on microbial fermentation for energy, with more than half of their maintenance energy requirement coming from microbial fermentation occurring in their enlarged caecum and colon. To achieve that, the gastro-intestinal tract (GIT) of horses harbors a broad range of various microorganisms, differing in each GIT segment, which are essential for efficient utilization of feed, especially to use nutrients that are not or little degraded by endogenous enzymes. In addition, like in other animal species, the GIT microbiota is in permanent interplay with the host’s cells and is involved in a lot of functions among which inflammation, immune homeostasis, and energy metabolism. As for other animals and humans, the horse gut microbiome is sensitive to diet, especially consumption of starch, fiber, and fat. Age, breeds, stress during competitions, transportation, and exercise may also impact the microbiome. Because of its size and its complexity, the equine GIT microbiota is prone to perturbations caused by external or internal stressors that may result in digestive diseases like gastric ulcer, diarrhea, colic, or colitis, and that are thought to be linked with systemic diseases like laminitis, equine metabolic syndrome or obesity. Thus, in this review we aim at understanding the common core microbiome -in terms of structure and function- in each segment of the GIT, as well as identifying potential microbial biomarkers of health or disease which are crucial to anticipate putative perturbations, optimize global practices and develop adapted nutritional strategies and personalized nutrition.

## 1. Characteristics of Horse GIT Microbiota along the Gastrointestinal Tract and Its Temporal Evolution during Life

Most of the information gathered in this review has been collected from publications where DNA sequencing methods or other Omics have been applied, as since the last 15 years those approaches have tremendously increased scientific knowledge on complex microbial ecosystems, in particular digestive ecosystems.

Microbiota refers to a complex ecological community of commensal, symbiotic and pathogenic microorganisms, including bacteria, archaea, fungi, and protozoa. Microbiome is, as recently defined [1] the combination of microbiota and its theatre of activity, which is composed by viruses, plasmids, extracellular DNA from dead cells, and microbial structural elements such as proteins, polysaccharides, nucleic acids, etc. Its composition is the result of long-term evolutionary adaptation of the host to its diet. In horses, gut microbiota has been mainly analyzed using DNA sequencing methods, targeting variable regions of the 16S rRNA gene as molecular markers for bacteria and archaea. Whereas eukaryotic communities such as anaerobic fungi and ciliate protozoa have been described in the equine hindgut and are indeed suspected to play a role in the digestion process [2,3], almost no published literature is available to date on these communities as assessed by DNA sequencing techniques so there is really a lack of information regarding their true ecological functions. Thus, this review will mostly focus on the microbiota composed by Archaea and Bacteria domains.

### 1.1. The “Common Core Microbiota” of the Adult Healthy Horse: A Myth or a Reality?

Because of its importance in healthy and sick horses, the GIT microbiota of horse should be looked with careful consideration of the digestive site. Richness and evenness increase towards the distal part of the GIT [4], indicating the complexity of this environment. A “common core microbiota” i.e., a group of microbial taxa that are shared by all or most horses [3,5] could exist in the various GIT segments, with strong differences highlighted between foregut and hindgut (Figure 1 and Figure 2). In the feces, the core community at the Operational Taxonomic Unit (OTU) level has been defined by “being present in all samples included in the study at 0.1% relative abundance (or greater)” [3]. Several studies report that fecal bacterial communities are not significantly different from the ones found in the colon [6], or even from the ones in the caecum [7], but differs from the upper tract. Thus, feces are interesting to be used as a non-invasive marker, although not complete, of what it happens in the hindgut (especially the colon) but not in the foregut.

In the foregut, transit time is relatively rapid, as digesta reach the caecum within 2–3 h of ingestion. Hence, only rapidly degradable (by the bacteria) and digestible (by the host) nutrients are used in that part of the GIT. The microbiota is dominated by Firmicutes and Proteobacteria (Figure 1) and the bacteria present here (including lactate-producing bacteria from Firmicutes) use starch, soluble sugars, readily fermentable fiber, and proteins or amino-acids. In line with that, putative functions of microbiota have been related to glycolysis, gluconeogenesis, pyruvate metabolism [8] and lactate and acetate are the main organic acids produced (Figure 1). The quantities can vary depending on diet type and sampling time [9]. If we detail the microbiota composition, *Lactobacillaceae*, *Streptococcaceae*, *Leuconostoccaceae* and *Sarcina* sp. (Clostridiaceae) in the Firmicutes phylum, and Pasteurellaceae (*Actinobacillus* sp.) in the Proteobacteria phylum are major bacteria in the stomach. Few *Prevotella* sp. and *Paraprevotella* sp. (Bacteroidetes phylum) and very few Fusobacteria are also detected. There is no strong difference between glandular and non-glandular regions [10,11]. *Actinobacillus* sp., *Lactobacillus* and *Streptococcus* are in majority tightly adhered to mucosal surface and are considered as part of the equine gastric mucosa [10,11,12]. Among *Lactobacillaceae*, strains of *Ligilactobacillus salivarius* and *L. agilis*, *Lactobacillus crispatus*, and *Limosilactobacillus reuteri* have been identified and are highly host specific [12].

One interesting observation is that whatever the study, there is a decrease in Bacteroidetes and an increase in Proteobacteria relative abundance from the stomach to the ileum (Figure 1), so that the common core microbiota in ileum is defined mainly by *Lactobacillaceae*, *Streptococcaceae* and *Pasteurellaceae* [13], with a strong presence of *Actinobacillus* sp. which may even become the most abundant taxa in the ileum. Proteobacteria have a role in maintaining microbial homeostasis through a constant dialogue with the host cells. They are facultative anaerobes, so they could play a key role in oxygen homeostasis, notably by consuming it [14,15], allowing the switch of the microbiota between the foregut and the hindgut. The regulation of oxygen concentrations by anaerobic facultative bacteria in the gut is associated with host control functions via oxidative phosphorylation coupled to fatty acid oxidation reactions in the mitochondria [15,16]. *Actinobacillus* sp. is part of the ileum microbiota of healthy pigs and humans, suggesting a high conservation between species. It may exert anti-inflammatory properties but is also a potential pathogen [17]. All those results highlight the importance of having a balanced microbiota in the foregut of horse, as a bloom of lactic acid bacteria or of Proteobacteria can be associated with disorders. The foregut microbiota varies strongly between horses, and the microbial composition in the stomach differs from the ones in the duodenum, jejunum and ileum [4]. The foregut is probably very sensitive and affected by the composition of the meal, the management, or other environmental factors, making this microbiota and the pre-cecal digestion process of high interest to understand genesis of dysbiosis and diseases in the total GIT.

In the hindgut, physiological conditions are different from the ones in the foregut, with a longer transit time, a lower pH (vs. small intestine) and redox potential, and substrates mainly composed by complex carbohydrates which have not been digested in the upper part of the GIT. The microbiota is here dominated by Firmicutes and Bacteroidetes (Figure 2). Other important phyla are *Verrucomicrobia*, Spirochaetes, Fibrobacteres and Actinobacteria. Proteobacteria and lactic acid bacteria are found at low relative abundance in the hindgut (Figure 2). Main Short Chain Fatty Acids (SCFAs) generated from dietary fermentation are acetate, propionate and butyrate and low concentrations of lactate (<1 mM). The microbial beta-diversity of the different hindgut segments has been found similar in several studies [8,10,13] whereas in other studies, caecum harbors a different community structure and metabolome when compared to colon [6]. Of note, results obtained in the various studies depend on the number of GIT segments considered, notably if the foregut was included [8,10,13] or not [6]. All in all, those results suggest that the bacterial communities are quite similar from one segment to another in the hindgut, making possible trying to design a common core microbiota although, due to some gradual changes and to the presence of the pelvic flexure, differences may appear from caecum to small colon. The number of bacterial Operational Taxonomic Units (OTUs) reported to be part of the core in the equine hindgut differs in the literature and is widely influenced by several factors linked to number of animals or diet composition, but also to DNA sequencing analysis pipeline (sequencing depth, application of a denoising step, rarefaction procedure). As a result, from 17 to 123 OTUs were identified [4,13,18,19,20]. Whatever the study, bacteria belonging to the order of *Clostridiales* (Firmicutes phylum) are part of the core, notably some unclassified *Clostridiales*, as well as *Lachnospiraceae* and *Ruminococcaceae* families [8,10,21]. It is not surprising to find those bacteria as core, as they are able to use complex carbohydrates, including cellulose, to produce SCFAs. As an example, *Lachnospiraceae* are well known butyrate producers in most mammal gut ecosystems and butyrate is known to have a protective function on colonocytes in the gut wall [22]. Interestingly, some OTUs belonging to these families have been correlated with expression of genes like *foxp3*, *IL-10* or *IL-17* [23], suggesting that, like in other animal species, some members of the *Clostridiales* order may have immune regulatory properties. Other important core families are Prevotellaceae, Paraprevotellaceae, and unclassified Bacteroidales within the Bacteroidetes phylum.

For obvious practical and ethical reasons, most studies on GIT microbiota composition have been performed on fecal samples. At the phylum level, Firmicutes are largely dominant among the fecal bacterial community, representing 15 to 85% of the total bacterial sequences (Figure 2). There are differences appearing for the next common phylum, which has been described to be either Bacteroidetes [24,25,26,27,28,29,30] or *Verrucomicrobia* [4,31,32,33,34]. In these studies, Firmicutes/Bacteroidetes ratio can widely vary. Due to the lower number of studies, such discrepancy is not yet well described in the hindgut, even if Dougal et al. [12] reported more *Verrucomicrobia* than Bacteroidetes in the colon. Methodological differences may contribute to explain those changes however it is unlikely the sole explanation, as in feces, those variations have been reported using various targeted 16S rRNA regions. Thus, if the feces microbiota accurately reflects the colon microbiota as stated in various studies, we can reasonably think that the switch between *Verrucomicrobia* and Bacteroidetes can also occur in the colon, and the functional importance of that remains to be elucidated. Other important phyla that are classically reported are Spirochaetes, Proteobacteria, Fibrobacteres and Actinobacteria. Archaea account for less than 0.5% in most of studies and are represented by the methanogenic genera *Methanobrevibacter* and *Methanocorpusculum* mainly. The low abundance of Archaea can be due to an underestimation related to the 16S rRNA regions used [35]. Fecal metabolome is logically dominated by short chain fatty acids, alcohols and ketones most likely arising from bacterial digestion of carbohydrates including dietary fiber [26]. Concentrations of SCFAs are slightly lower than those found in the hindgut (Figure 2).

The mucosal microbiota of the GIT is also of clinical interest as these bacteria are in close contact with the host cells, and, by this way, may influence the host physiology. Unfortunately, very few studies exist on the mucosal microbiome [10]. The same phyla as those reported for the luminal contents are described in the mucosa: Firmicutes, Bacteroidetes, *Verrucomicrobia*, Tenericutes, Spirochaetes and Fibrobacteres. At the OTU level, *Lactobacillus* sp. and *Actinobacillus* sp. are dominant taxa in the mucosa of the stomach as in the lumen, but higher bacterial richness is found in mucosa [10]. In hindgut mucosa, OTUs are very similar to the ones found in the lumen, except for a higher relative abundance of *Desulfovibrio* sp. (Proteobacteria phylum) and a reduction of *Treponema* sp. on the cecal segment. Interestingly, Lindenberg et al. (2019a) found a correlation between the Desulfovibrionaceae family and the expression of some immune genes in the cecum and colon of horses, suggesting an active role of those bacteria in immune regulation [21]. In addition, in ventral and dorsal colonic mucosa, more Archaea (*Methanobrevibacter* and *Methanocorpusculum*) were found than in the lumen. More attention would have to be paid to the Archaea as they are present in biofilms, with close interactions with the host cells and other bacteria (hydrogen transfer). A lower concentration of methane has been measured in horses compared to ruminants (92 ± 15 and 28 ± 9 L/kg digested neutral detergent fibre for ruminants and horses respectively, [36]. An elegant hypothesis to explain this difference is that obligatory H_2_ production during forage fermentation is captured in CH_4_ in the ruminant where ruminal gases are readily released by eructation, while in acetate in the equid hindgut where a build-up in gas pressure could potentially damage these organs. Another hypothesis could be that the CO_2_-reducing acetogenic bacteria could be more competitive in horses than in ruminants.

The common core microbiota in equines is still under discussion, and, based on our investigations, some OTUs, not considered to be part of the common core microbiota *per se* in the above cited studies need to be carefully described in further research. First, recent investigations showed that the genus *Akkermensia* was quite abundant in the hindgut or feces of horses [4,8,21,27,29]. This bacterium is of high interest in humans as it would have anti-inflammatory properties. Its abundance is negatively correlated with the incidence of obesity, diabetes, or metabolic disorders. In agreement with the findings in humans, Lindenberg et al. found a positive relationship between the relative abundance of *Akkermansia* sp. in the ileum and the expression of *foxp3* in the mesenteric lymph nodes, suggesting a role in the oral tolerance to commensal bacteria [23]. Other interesting taxa located in the hindgut that appeared in various studies are affiliated to *Treponema* sp. (Spirochaetes phylum; Figure 1). In the study of Daly et al. [37], most of the species cloned with *T. bryantii* and *T. succinifaciens*, both being not considered as pathogenic. *T. bryantii* uses fermentable substrates, in particular soluble sugars from cellulose degradation by *F. succinogenes* for example. *T. succinifaciens* is strictly saccharolytic and produces large amounts of succinate. *F. succinogenes*, a fibrolytic bacterium, also appears quite consistently in the gut microbiota of horses fed with hay or grass, but not with high concentrate diet [19,30,33]. It is a widely described bacterium in ruminants and the evolution of abundance of the related OTUs could be interesting, as it has been described in ruminants as a biomarker of the fibrolytic activity in the rumen, with the strong decrease in case of ruminal acidosis. Lastly, *Phascolarctobacterium* sp., which uses succinate to produce acetate and propionate, is also found in several studies using horses. This bacterium was reported to positively correlate with maintenance of normal weight in children [38] and positive mood in adults [39] and to lower levels of liver triglycerides following a high-fat diet in a nonalcoholic fatty liver rat model [40]. In horses, little is known about the exact role of this bacterium, but its relative abundance is reduced in intestinal microbiota from oligofructose-induced laminitis when compared to those from healthy horses, together with a lower relative abundance of *Akkermansia* (and *RFP12*) and *Fibrobacter* [41]. Furthermore, Edwards et al. [42] pointed out all those OTUs as being part of the common core microbiota of the hindgut of equines when considered several sub-species (i.e., horses, donkeys, zebra).

The composition of a “normal” GIT microbiota and the “common core microbiota” are still difficult to define in horses, as the definition varies from one study to another. One interesting thing we must pay attention to is that in opposite to ruminants and other types of animals where a quite limited number of OTUs is highly dominant, the microbial core community of horses is not dominated by any particular OTU. Although we start to have an idea on the common core microbiota of horses, there is still a big proportion of unclassified read assignments at the genus level in large intestine and feces, suggesting that those samples may contain genera that are distinct from those isolated from other mammals. We clearly lack cultured representatives of those novel bacteria, as only 30% cecal microbiota have been cultivated [43]. It will be important to work in the future to well-establish and understand the GIT microbiota of the horse. We also lack information about functions of the microbiota, i.e., the “functional core microbiota”.

In addition, methodological differences in DNA extraction protocol, type of sequencing platform, selected region of the 16S rRNA gene and type of corresponding primers may play a role in the discrepancies that can be seen between results [3,35]. Alterations of fecal microbiota may appear, and Stewart et al. (2018) demonstrated that bacterial community varies significantly between center and surface of the fecal balls, while being stable from 0 to 6 h after defecation. In addition to these methodological biases, geographic locations of the studies, seasons, breed-specific differences, age, diet composition, variations in management and feeding conditions can impact microbiota composition. A recent study [44] reported differences in composition and diversity between different sport breeds (Hanovrian, Lusitano, Arabian and central European breeds), with 27 genera varying in abundance across breeds. However, a non-significant correlation was observed between microbial composition and the host pedigree-based kinship, allowing authors to conclude that breed exerted only limited effects on the equine fecal microbiota. In another experiment, Zhao et al. [29] attempted to discriminate the microbiota of Mongolian vs. Thoroughbred horses, however, they were fed differently making conclusions impossible to draw. Variation associated with seasonality and change in forage type occur over a 12-month period [45,46].

### 1.2. Evolution of the Horse Microbiota with Age

The foal’s first week of life is considered a critical period, with increased morbidity and mortality due to respiratory diseases, enteritis and sepsis. Therefore, optimizing the colonization of the foal’s gut from birth, with a good balance of functionally important microbial communities is an important objective both at nutritional and health point of view.

The role of the mare is key in the acquisition of gut colonization process in the newborn as microbial transmission occurs thanks to repeated contacts between neonate and its dam. Indeed, foal’s meconium contains a diverse bacterial community, composed by *Acinetobacter*, *Stenotrophomonas* or *Sanguibacter* reported as opportunistic bacteria from several animal hosts, *Aerococcus* (from *Clostridiales* order), an anaerobic fermenter of mammalian gut ecosystem, and other common gut microbiota members, such as *Streptococcus, Enterococcus*, and Enterobacterioaceae [45]. As the composition of the mare gut microbiota was shown to be close to that of the rectal microbiota of its foal at birth [14], dam’s gut microbiota seems to specifically contribute to the meconium community by providing microbial components from the gut ecosystem. In addition, amniotic fluid contains a low bacterial DNA load, which would represent a source of heterogeneous microbial subset to the fetus [45]. Thus, as it has been also hypothesized in other animal species and humans, internal transmission route of microbial components could occur through the mare’s dendritic cells, which penetrate the host intestinal epithelia, sampling luminal bacteria or bacterial antigens that would be then released into the placenta via the bloodstream [47,48,49]. The delivery of bacterial components through this route to the foal would prime the newborn’s immune system for further shaping of the microbiota in the gut. However, these recent data having been obtained using DNA sequencing-based approaches, on quite low biomass samples, it is still unclear whether this microbial transfer is under the form of live, dead, or fragmented bacteria.

From birth, the gut microbiota of the foal progressively evolves to reach a diverse and functional microbiota, and it has been shown that after 2 months of age, the foal’s gastrointestinal microbiota has been established to include the bacteria necessary for the digestion of the roughage typically found in the mature horse diet [50]. However, until this age, the gut microbiota is subjected to drastic changes in abundance and composition (Figure 3). Abundance of fecal bacteria strongly increases from birth to 24 h of life to reach above 10 billion of 16S gene copies per fecal swab at 7 days of age [51]. A higher bacterial richness has often been reported at birth than afterwards, which may result from environmental exposure, as the neonatal foal encounters a wide range of maternal and environmental bacteria that are not true colonizers, but only transient organisms [14,16]. In the early period, the initial microbiota is indeed highly dynamic and unstable, with a large inter-individual composition [21]. As at birth the redox potential in the hindgut is positive, environmental conditions are favorable for the maintenance of aerobes and facultative anaerobes belonging to Proteobacteria (*Escherichia/Shigella*, *Acinetobacter*), or Firmicutes (*Staphylococcus*, *Streptococcus*, *Bacillus*, *Lactobacillus*). In fact, at one day of age, several genera commonly observed as cause for neonatal foal sepsis are prevalent [51]. This could represent a kind of reservoir for opportunistic pathogens, which could emerge in case of dysbiosis, or on the contrary, the presence of these commensal bacteria in their ecological niche in a well-balanced ecosystem would prevent newborns to be further colonized by pathogenic members from the same genera, as suggested in piglets [52,53].

The behavior of the foal may play an important role on the colonization process. Indeed, the first episode of coprophagy, occurring generally at 3–5 days of age, is related to rapid changes in fecal bacterial composition, with the acquisition of SCFA producers, such as *Prevotella*, *Blautia* or *Ruminococcus* which belong to the core gut microbiota in adult [45].

Seven days after birth, the structure of the fecal bacterial microbiota of foals is closer to that of the mare’s feces, but still different [51]. At this age, Firmicutes are dominant, Proteobacteria are progressively replaced by Bacteroidetes and Fusobacteria (Figure 3). The most abundant genera are *Bacteroides*, *Fusobacterium*, *Tyzzerella*, *Streptococcus* and *Lactobacillus*, the genus *Akkermansia*, tightly linked to the mucosal phase of the gut, was also found in a majority of animals in a study by Husso et al. [51]. Looking at the dam’s microbiotas at different sites (vagina, oral, fecal), the genus *Bacteroides* was almost exclusively found in the vagina of the mares, which confirmed that the vaginal ecosystem is an important source of microbes, as also demonstrated for cattle [54], but also that maternal imprinting was still measurable after one week of age. The strong increase in *Bacteroides* can be linked to the start of solid food intake, which generally occurs by one week of age, through dam’s forage and concentrate [50], and the families *Lachnospiraceae* and *Ruminococcaceae* are detected from this age at a significant abundance. Mare’s milk has also a strong impact on the microbial composition in the foal’s gut, with possible transmission of *Lactobacillus* and *Streptococcus* representatives, but also through a promoting effect of oligosaccharides that act as prebiotics towards certain bacterial species [21]. With the start of anaerobic fermentation process, the gut ecosystem matures to progressively reach a high potential for degradation of the dietary compounds offered to the young foal at weaning.

Weaning is undoubtedly a very stressful event in the life of horses. Indeed, at weaning profound changes occur in feed composition, feeding and social behavior, as well as mare-foal interactions, which put the young foal under stress and may affect gut microbiota balance. Besides a direct effect of the modification in feed composition and intake, which directly impacts microbial structure and activity as well as microbiota relationship with the host gut mucosa, weaning stress induces the activation of the Hypothalamic-Pituitary-Adrenal (HPA) axis which releases catecholamines and glucocorticoids both at the intestinal and the systemic level. The gut response to these stress hormones is the synthesis of cytokines, neurotransmitters and hormones that may modify microbiota diversity, activity and may promote intestinal pathogens [55,56] and in the other way round, gut microbiota metabolites may also regulate the stress response [57]. Mach et al. evaluated the impact of maternal separation and of two different weaning methods (progressive vs. abrupt) on the composition of gut microbiota with DNA sequencing approach, and investigated how the shifts observed post-weaning could affect the host, in particular growth, salivary cortisol and blood telomere length [58]. Regardless of the weaning method, maternal separation at weaning markedly shifted the composition of gut microbiota in all foals, with three distinct communities observed 3 days post-weaning. These community types were related to different host responses to stress with also different consequences on host physiology. It was also reported that the relative abundance of genera belonging to Prevotellaceae family and *Ruminococcus* genus, generally associated with beneficial functions, was lower with progressive weaning compared to abrupt weaning. Of note, during progressive weaning, anaerobic fungal load was increased, which was positive owing the capacity of these microbial population to hydrolyze plant cell wall polysaccharides.

Beside early life, as the longevity of horses increases, the microbiota may evolve with age as it has been described in other species. Indeed, in humans and in dogs, it is admitted that senescence and inflamm-aging are related to changes in fecal microbiome. Data on the effect of age on microbiota in horses are still scarce. Dougal et al. [19] showed a decrease in richness, but no difference in the structure and composition of the fecal microbiota in healthy old horses (19–28 y old) vs. adult horses (5–12 y old); while McKinney et al. [34] observed no difference between health old (median 22.6 ± 1.8 y old) and adult horses (median 6.8 ± 3.7 y old). On the contrary, Morrison et al. [18] reported a greater alpha-diversity and higher relative abundance of Proteobacteria, and lower relative abundance of Fibrobacteres in old (mean 21.55 ± 2.94 y old) vs. adult ponies (mean 9.83 ± 3.21 y old), speculating a better capacity to use fiber in adult than in old equine. However, despite the elegance of the study, the authors failed to find clear biomarkers in either fecal microbiota or metabolome, underlining the difficulty to work only on feces, not indicative of the complete digestive tract, to make relationships between GIT microbiota and host phenotype.

## 2. Horse GIT Microbiota Changes According to Abiotic Factors

### 2.1. Impact of Diet Composition

In equines as in other mammals, diet is a key factor influencing the composition of gut microbiota. Horses are physiologically well adapted to digest high forage diets, thanks to the presence of fiber-degrading bacterial [18] and fungal [59] active communities in their hindgut. However, this forage diet is often replaced by high-starch diets to increase energy density which is generally required to exercised horses. Several studies have compared the impact of a high starch vs. high forage diet on fecal microbiota in horses (Table 1). The studies conducted by Medina et al. [60] and Jouany et al. [61] have focused on viable cultivable functional bacterial groups and were the first to demonstrate a shift in bacterial populations with a high starch diet. These bacterial changes were associated with reduced polysaccharide-degrading and glycoside hydrolase bacterial activities due to lower pH and increase in lactic acid concentration and impaired fiber digestibility. In ruminants, this type of dysbiosis is well characterized in the rumen as subacute ruminal acidosis (SARA) [62]. In equines, disruptions of the composition and/or activity of hindgut microbiota lead to altered digestive health. Indeed, the accumulation of lactate and the reduction in pH leading to subclinical acidosis in the caecum-colon increase the horses’ susceptibility to colic pain or laminitis [63]. Lactate-producing Gram positive bacteria such as *Streptococcus bovis*-*Streptococcus equinus* species have been identified as being involved in the onset of laminitis in horses fed an oligofructose load [64]. Interestingly, even small amounts of starch added in the diet can result in hindgut dysbiosis, as shown by an increase in the abundance of an OTU related to lactate producing *Streptococcus* with less than 1g of starch per kg BW [65].

In the study of Morrison et al. [67], 23 ponies were followed during a 2 years experimental period and the effect of transition from high forage to high starch diet was investigated. Results confirmed that fecal pH was decreased, and there was a significant increase in the relative abundance of *Candidatus*, *Saccharibacteria* and *Firmicutes* together with a reduction in the relative abundance of the fibrolytic bacterial phylum *Fibrobacteres* across the study days, following the addition of barley to the diet. Of note, two categories of age could be studied but data showed that age had a minimal influence on the microbiota response to the diet. Furthermore, at the genus level, an increase in *Streptococcus* abundance was noticed but only in some individuals. In these animals, identified as ‘*Streptococcus* responders’, gut ecological conditions were suspected to be particularly favorable to *Streptococcus* overgrowth. The bacterial alpha-diversity was significantly decreased, which would identify those animals as particularly at risk for hindgut acidosis or laminitis. It is thus of importance to analyze the data at the individual level to be able to identify peculiarities in gut microbiota associated to an exacerbated animal susceptibility towards gastrointestinal disorders. The addition of a cereal starch supplement to a hay diet to mares has shown to decrease the core bacterial community which may increase the risk for subsequent metabolic dysfunction [19].

Feeding high starch diets not only alters gut microbiota composition, microbial activities, hindgut environment and digestion capacities, behavioral reactivity would be also affected [65,68,69]. Feeding starch to ponies led to increased frequency of pace-change and to decrease in the time spent investigating their surroundings, and this behavior was strongly associated with a change in fecal microbial profile compared with high forage diet [65]. Generally, more reactive behaviors make horse less predictable and thus more difficult to handle. A positive correlation was found between the frequency of blowing after a novelty test with the relative abundance of Succinivibrionaceae family, and with the concentration of amylolytic bacteria in the colon of high starch fed horses [68]. With a cohort of 185 healthy horses encompassing a large range of age, sex, athletic disciplines and performance, Mach et al. [70] showed associations between gut microbiota composition and behavior reflecting welfare deterioration. These new data are really promising but more research needs to be performed to better understand which microbial signals are responsible for this dialogue with the nervous system, to find new avenues for prevention of horses’ compromised welfare states in stressful events.

Besides diet composition, feeding management, such as meal size and frequency, also affects gut microbiota. Indeed, horses fed three small meals had a different cecal microbiota than horses receiving a single large meal per day, with a lower relative abundance of *Prevotella*, *Lactobacillus*, *Streptococcus*, *Coprococcus* and *Phascolarctobacterium* [71].

### 2.2. Transportation

Transportation stress is recognized to be a major contributor in many health related issues in horses, among which gastro-intestinal disorders play an important part [72]. Several factors may significantly impact gut health during transportation, such as humidity, temperature, air quality, feed, and water allowance. An Australian survey on transport management practices reported that gastro-intestinal disorders represented ~30% of the transport-related health problems [73]. Because of gut microbiota-brain axis communication system can be altered in stressful situation, transportation stress may negatively affect hindgut microbiota composition and activities. Indeed, a 2 h-truck transportation could already trigger a change in fecal bacteria populations and induce a dysbiotic state [74]. Perry et al. performed a study where travel cecum-cannulated horses were transported to an unfamiliar location, stalled to simulate horse show condition, and returned to the equine center [72]. Compared to control horses which stayed in the equine center, alpha-diversity of cecal microbiota of transported horses was decreased. There was a significant decrease in Bacteroidetes phylum relative abundance for travel horses during the transportation and return phases, compared to baseline. Several other taxa were also affected, such as lactate producers such as *Lactobacillus* and *Streptococcus* whose relative abundance increased, whereas taxa known to be involved in depolymerisation of complex dietary carbohydrates, such as *Ruminococcaceae* and *Lachnospiraceae* families, decreased.

### 2.3. High Performance

In endurance horses, physical and psychological stresses may also affect gut microbiota-brain axis. In mice exposed to 6 weeks of forced treadmill running, physical and emotional stress during exercise was highly correlated with changes in gastrointestinal microbiota composition, with for instance changes in the balance of bacterial populations involved in intestinal mucus degradation and in immune function [75]. During high intensity exercise, the redistribution of blood flow away from the intestines, together with thermal damage to the intestinal mucosa can cause intestinal barrier dysfunction, followed by an inflammatory response and leaky gut [76]. In the study of Mach et al. [70] with a cohort of 185 healthy horses classified according to equitation discipline, specialty and level of performance, gut microbiota composition was associated to equitation conditions; in particular, Gala and Cadre Noir specialties linked the most to fecal microbiota composition, notably with a decrease in relative abundance of *Lachnospiraceae* AC2044 group and *Clostridiales* family XIII, both groups being butyrate-producers, probably due to very high physical and mental stress during training and show events, compared with other specialties. Mach et al. [77] studied the gut-mitochondria crosstalk in endurance horses and described a subset of mitochondria-related differentially expressed genes involved in pathways such as energy metabolism, oxidative stress and inflammation in sportive vs. resting horses. Interestingly, these genes were associated with butyrate-producing bacteria of the *Lachnospiraceae* family, especially *Eubacterium*. Microbiota modulation therefore appears to be a potential strategy to enhance athletic performance.

### 2.4. Heat Stress Conditions

According to Cymbaluk and Christison in their review [78], the five climatic variables of a horse’s microclimate are ambient temperature, relative humidity, precipitation, wind velocity and solar radiation, and the most important single climatic stressor is ambient temperature. Feed intake by horses can decrease by 15 to 20% under high temperatures [78]. In ruminants as well, an increase in temperature and humidity index (THI) was reported to trigger a decreased intake. Under heat stress, rumen motility decreases and thus feed has a prolonged residence time within the rumen, which leads to increase dry matter digestibility [79], and to changes in microbiota community composition and metabolic activity [80] leading to a higher risk of ruminal acidosis and a lower milk performance [81]. Uyeno et al. [82] reported for example that the relative populations of the *Clostridium coccoides*-*Eubacterium rectale* group and the genus *Streptococcus* increased, and that of the genus *Fibrobacter* decreased in response to increasing temperature. In horses, dry matter digestibility has been reported to increase as well with high THI [78], thus it can be hypothesized that hindgut microbiota would also be negatively affected and that a dysbiotic state is triggered under heat stress conditions in equines. In monogastric farm animals (swine and poultry), gut microbiota compositional dynamics have been suggested to be further investigated as biomarkers of heat stress [83] and in pigs, the abundance of *Ruminococcus bromii* before heat stress has been significantly positively correlated with energy retention during heat stress [84]. Energy retention being generally decreased under heat stress, these results suggest that this species would be a keystone marker of heat stress susceptibility.

## 3. Horse GIT Microbiota and Digestive Disorders: What Do we Know?

### 3.1. Equine Gastric Ulcer Syndrom (EGUS)

Gastrointestinal disorders are a common cause of diseases and death in horses, and modulation of the gut microbiome have been observed in various cases. In the foregut, EGUS is a major disease and may affect 53 to 90% of adult horses. It is characterized by the ulceration of the mucosa of the esophagus, stomach or duodenum. Dietary management, stress, nonsteroidal anti-inflammatory drugs, excessive performance are associated with high prevalence of EGUS. Most of the time, it is associated with appetite losses, abdominal pain, poor body condition and decreased performance. Two types of EGUS may appear, equine glandular gastric disease (EGGD) and equine squamous gastric disease (ESGD) that differ in pathogenesis. ESGD affects the squamous part of the stomach because of pH reduction due to higher hydrochloric acid secretion and high fermentation rate in horses fed with grain-based diets (more than 2 g/kg BW starch/meal), combined with impairment of mucosa integrity. EGGD would be rather associated with decreased mucosal defenses, but its etiology is still unknown.

The change in stomach microbiota is not clear. Overall, very little is known about the relationships between gastric ulcer and microbiota, as no bloom of streptococci or lactobacilli or other bacteria has been clearly observed. Contrary to what it has been observed in many species including humans, there are only weak evident relationships between gastric ulcer and *Helicobacter* infection [11], although Dong et al. [85] found the presence of *Helicobacter* in the horse stomach using in silico approach, but not culture or PCR techniques. A recent study [86] showed that neither Firmicutes nor Proteobacteria OTUs were enriched in the stomach of horses suffering from EGGD. *Helicobacter* sp. was detected but again, not in association with gastric ulcer. Modest differences in the community structure of the gastric glandular mucosal microbiota among EGGD and healthy horses using Jaccard similarity index (index to define beta-diversity) were obtained, and the authors speculated that the presence or absence of specific bacteria might be associated with EGGD, rather than a clear dysbiosis. Dong et al. [85] analyzed microbial communities of the glandular region using 10 thoroughbred racehorses healthy or having moderate to severe gastric ulcers and reported similar alpha-diversity, same common core microbiota, but increase in various proportions of OTUs like *Clostridium_g19* and Staphylococcaceae_uc and decrease in *Dietzia cinnamea* in horses suffering from mild to severe gastric ulcer. However, the relative abundance was very low, making difficult to draw any conclusion. Horses fed with high concentrate diets had higher relative abundance of Firmicutes without being related to the severity of the gastric ulcer, maybe due to the low number of animals used. On the contrary, Voss et al. [87] reported a higher relative abundance of Firmicutes in samples collected from EGGD lesions due to a particularly high relative abundance of *Sarcina* (up to 92.4%) in two horses with EGGD. *Sarcina* has been identified in histopathological samples from humans with gastric disease while *S. ventriculi* has been associated with diseases in other animal species. Future research is required to confirm if *Sarcina* sp., although not causative, could be used as a possible marker of functionally or structurally delayed gastric emptying. The activity of microbes as well as the interplay between microbes and host cells could change rather than the proportion of microbes, and maybe the use of techniques with higher resolution, and notably whole genome sequencing, functional metagenomics or metatranscriptomics, combined with metabolomics is required to obtain more complete understanding of the community and function of the stomach microbiota of horses suffering from EGUS, and more specifically of EGGD.

### 3.2. Colitis and Diarrhea

Colitis constitutes a leading cause of critical illness in horses and can be accompanied by increased risks for severe complications. The specific cause of colitis remains unknown in more than 50% of cases while traditionally, *Clostridium difficile*, enterotoxigenic *C. perfringens* and *Salmonella* spp. have been incriminated as the most important etiological agents causing diarrhea in horses. Other factors like sand impaction, antibiotic treatment, carbohydrate overload can also be involved. Disruption of the normal microbiota is likely a key factor in most cases of colitis and diarrhea, and several studies aimed at describing the microbiota of healthy and sick horses under different conditions (Table 2). Several studies indicate that the equine fecal microbiome of healthy horses has a significantly greater alpha-diversity compared to horses with colitis [27,34]. On the contrary, Costa et al. [30] and Arroyo et al. [88] reported no change in richness of the feces or colon microbiota between colitis and healthy horses; however, the later observed a greater richness in the colon mucosa vs. lumen of colitis horses, underlining once again that the alpha-diversity index, although often considered as a biomarker of a healthy microbiota, is not sufficient *per se*. If the results obtained on alpha-diversity are not consensual, most of the authors report a clear effect of colitis on beta-diversity of cecal, colon and fecal microbiota [27,30,34,88] suggesting that the microbiota structure of horses with colitis is significantly different to that of healthy horses. The comparison of community composition between mucosal and luminal content revealed differences in both the cecum and colon microbiota of colitis cases, but not in cecum and colonic microbiota of healthy animals [88].

When going deeper into the bacterial composition, different bacterial taxa can be pinpointed as being significantly different between healthy and colitis horses. McKinney et al. [34] reported that the microbiota of diarrheic horses was characterized by a low abundance of *Verrucomicrobia* and of Fibrobacteres, and the fecal microbiota transplantation they applied resulted in the increase in alpha-diversity index, and the relative abundance of *Verrucomicrobia* and in the decrease in Proteobacteria. Costa et al. [30] reported that the proportion of Firmicutes (68% vs. 30% in healthy vs. diarrheic horses) and Bacteroidetes (14% vs. 40% in healthy vs. diarrheic horses) tended to be different. In addition, a significant increase in the relative abundance of Fusobacteria has been reported in several studies [27,30]. Rodriguez et al. [27] observed greater relative abundance of *Actinobacillus*, *Porphyromonas*, and RC9 gut group, while having lower relative abundance of *Roseburia* and of a taxonomically undefined population belonging to *Ruminococcaceae* family. These authors tried to discriminate main potential bacterial biomarkers by creating different diagnosis categories, among which diarrhea without other symptoms, and they revealed that *Akkermansia*, *Fusobacterium*, *Porphyromonas* and *Xylanibacter* genera were typical from the diarrhea group. Even if the number of horses included in the study was low, this approach is quite interesting to understand the microbial dysbiosis markers involved in different pathologies. In mucosal and luminal contents of both colon and cecum, members of *Lactobacillus* spp. were strongly associated with colitis, as well as *Escherichia* and *Fusobacterium* spp., while members of *Lachnospiraceae* family and *Fibrobacter* spp. were associated with healthy horses [88].

Normal microbial inhabitants of the gut are thought to help maintain a balance between inflammatory and anti-inflammatory mediators in the intestinal tract. The crosstalk between host cells and bacteria is thus crucial. In case of colitis, gut inflammation caused by pathogens or other factors may significantly alter the gut environment which, in turn, impacts the microbiota balance in the GIT, re-shaping the resident microbial community, as suggested by the huge change in beta-diversity. In addition, inflammation associated with colitis may result in the increase of certain nutrients that may selectively promote the growth of potential pathogenic bacteria. Of note, most of bacteria reported here as characterizing the microbiota of colitis horses are bacteria with strong pro-inflammatory lipopolysaccharides and have the capacity to use host-derived inflammatory by-products (e.g., nitrate) as energy sources. Microbiota of colitis horses is depleted of certain bacterial members known as SCFA producers, like *Lachnospiraceae* or *Ruminococcaceae* families, or of key fiber-degrading bacteria like Fibrobacteres, highly pH and oxygen sensitive. A decrease in SCFA production may increase oxygen respiration by the epithelial cells, increasing in turn the level of oxygen in the lumen, favoring the growth of facultative anaerobes like *E. coli*, *Actinobacillus*. Overall, in case of colitis, differentiating cause and effect is not possible without a greater understanding of pathophysiology, but identification of organisms disproportionately present/absent in horses with colitis strongly suggests an important crosstalk between host cells and microbiota as shown in other species. In mice, during *Salmonella enterica*-induced colitis, luminal oxygen availability increases, as indicated by an oxygen respiration-dependent bloom of the pathogen in the colon and a concomitant decline in the abundance of obligate anaerobe Clostridia [95]. Further research to improve our understanding of the inflammatory pathways that interact with bacteria may elucidate reasons behind varying presentations of the same disease and various responses to the same treatment in different individuals. Furthermore, it could allow developing new adapted and individual nutritional strategies (pre-, pro- and post-biotics, see Section 6) to modulate the physico-chemical parameters of the GIT to, in turn, re-shape the microbial community.

### 3.3. Colic

Colic is an important disease in equines, however, once again, the etiopathology remains not fully understood. With growing evidences of the crucial role of the GIT microbiota and its interplay with host cells, recent studies have begun to evaluate its composition in horses suffering from colic. While beta-diversity remains unchanged, alpha-diversity of the microbiota has been reported either as being significantly decreased [89,90], or not affected [32,91]; Table 2 in horses suffering from colic compared to healthy subjects. A study using 12 horses suffering from colic, among other suffering from other gastrointestinal diseases [27] highlighted a higher abundance of *Escherichia* and *Streptococcus* genera in colic group. In their recent study, Park et al. [89] reported a change in the fecal Firmicutes/Bacteroidetes (F/B) ratio, which increased in horses with an intestinal disease compared to healthy controls, in agreement with previous reports [32,89]. More precisely, the average F/B ratios were 1.94, 2.37, and 1.74 for horses with large intestinal disease, small intestinal disease, and healthy controls, respectively. However, the F/B ratio alone is not sufficient to evaluate the disease status, as this ratio is not systematically significantly altered. At the expense of this ratio, Weese et al. [32] proposed that the Firmicutes/Proteobacteria ratio could be of interest, as they observed a decrease in the relative abundance of Firmicutes while an increase in the one of Proteobacteria.

If we go more precisely, the different studies performed to evaluate the change in fecal microbiota did not report consistent results (Table 2). The decrease in relative abundance of fibrolytic bacteria like Fibrobacteres [90] and *Ruminococcus* [32,91] is not observed in all studies. However, the decrease in Methanobacteriaceae family (including *Methanobrevibacter*) observed in horses with large and small intestinal colic compared to healthy horses [95] is interesting to note because those Archaea are pH sensitive and are associated with an active fibrolytic activity in the colon. In several studies, an increase in relative abundance of bacteria belonging to Proteobacteria, was observed notably in potentially opportunistic pathogens *Acinetobacter* and *E. coli*/*Shigella* [27,32,89,90]. An increase in relative abundance of *Lactobacillaceae*, *Streptococcus* and *Bifidobacterium* has also been noticed in some studies [27,89]. This observation may support previous findings that excessive lactate production and decrease in the hindgut luminal pH are associated with an increased relative abundance of lactic acid bacteria in horses with colic. Although the beneficial effects of *Lactobacillus* and *Bifidobacterium* are well documented in humans, such a result may suggest that the role of specific microbes can vary in different animal species, according to their abundance and the nature of interrelationships developed within the microbial network.

Overall, literature is still scarce, and a lot of factors increase the variability of the results (type of colic (large/small intestine), type of sample (feces or colon) and timing of sample collection, heterogeneity of the studied population, low number of animals, targeted 16S DNA region sequenced). However, some interesting key bacteria can be underlined. Notably, colic seems associated with a disruption of the methanogenic and fibrolytic activity, and associated with increasing relative abundance of Proteobacteria and of certain lactate-producing bacteria. This may suggest that colic is related with higher lactate production, resulting in decreasing pH and fibrolytic bacteria activity, and in turn to lower SCFA production, leading to a change in the crosstalk with colonocytes and, finally, making possible the growth of facultative anaerobes like certain Proteobacteria. Once again, further studies are required to understand the pathophysiology of this digestive disorder and the existing crosstalk between host cells and microbiota.

### 3.4. Free Fecal Water

A definition of Free Fecal Water (FFW) also named Free Fecal Liquid or Fecal Water Syndrome is proposed by Kienzle et al. [96] as the condition in which horses produce normal feces, but before, after or during defecation, fecal water runs out of the anus. Usually, no effects on general health and welfare are reported, so FFW is not considered as a true pathology. The role of gut microbiota as a factor involved in FFW is still currently speculated as no clear difference in microbiota alpha-diversity and taxonomic composition has been reported between FFW and healthy horses so far [92,93,94] (Table 2). However, fecal microbial transplantation (FMT) has been proven successful to decrease FFW symptom severity, even if no modification in fecal microbiota of FFW horses who positively responded to FMT [93]. Of note, the fecal transplant contains not only microbes but their associated metabolites, and those could be important in the observed benefit. In addition, microbiota composition has been analyzed in the solid phases of the feces, but no analysis has been performed in the liquid phase. To increase our understanding, metabolomics and microbiota analyses of both liquid and solid phase of the feces have to be done.

### 3.5. Parasitism

Intestinal parasites are often considered as a major threat for equine health by the direct interaction with gut microbiota and epithelial host cells [97]. One third of chronic diarrhea has been attributable to both large and small strongyles, especially cyathostomins larvae. Young horses are very sensitive to intestinal parasite particularly due to an unestablished immunity. The French Network responsible for the Equine Disease Prevention (RESPE) reports that 70% of foals and weanlings (6–24 months) will be contaminated by at least one parasite. The presence of helminths in the intestinal lumen may alter the gut microbiota and composition, as reviewed by Midha et al. [98] and Peachey et al. [99]. While parasite infection seems to have various impacts on bacterial diversity [98], the Methanomicrobia class of Archaea methanogens was found negatively correlated with the fecal parasitic load and an increased abundance of Proteobacteria phylum was observed in contaminated horses [100]. Although not reported for equines, in mice a consistent finding across various helminth infections is an increased abundance of *Lactobacillaceae* [98]. Differences in abundance of several bacterial taxa and changes in fungal and protozoal loads were found in the feces of ponies that were either susceptible or naturally resistant to parasitic infections [97].

A strong inter-individual response of horses is observed in front of helminth infection, notably with the presence of resistant phenotype. The high prevalence of butyrate producers in the microbiota would, at least in part, explain the resistant phenotype, as butyrate is recognized as an anti-inflammatory compound. *Clostridium* cluster XIVa has also been shown to prevent the growth of opportunistic pathogens such as *Campylobacter* or *Pseudomonas*, which could participate in the global pathophysiology of infection. Moreover, in susceptible ponies, gut microbiota alterations would lead to changes in several immunological pathways such ad pathogen sensing, lipid metabolism, and activation of signal transduction that are critical for the regulation of immune system and energy homeostasis. In addition, as gut of parasites may harbor microbial communities taken from their environments, they can be considered as vectors of potential pathogens that could be released at the mucosal level of the infected host [98].

Anthelmintic drugs (benzimidazoles, tetrahydropyrimidines and macrocyclic lactones) are commonly used in routine to control parasite load, and besides resistance development issues that are more and more widespread [101], these compounds may alter gut microbiota. Goachet et al. [102] showed that gut cellulolytic bacteria were negatively affected by the administration of moxidectin and a decrease in alpha-diversity together with a significant change in 21 bacterial OTUs relative abundance were reported after the use of moxidectin and praziquantel deworming solutions [103]. Our group studied the impact of a deworming administration of Praziquantel and Ivermectine on 8 horses differing in gender, age and breed, and we showed that a significant reduction of the ratio F/B in all horses [104]. In addition, we showed a significant negative correlation between Shannon diversity index and the Proteobacteria/Fibrobacteres ratio (Figure 4). The feces of 2 ponies exhibited an increase in Proteobacteria and a decrease in Fibrobacteres relative abundances just after deworming, suggesting that on top of the global trend in modulating the F/B ratio, there is individual pattern in response of a deworming drug.

## 4. Systemic Disorders: Laminitis, Equine Metabolic Syndrome (EMS) and Obesity and Their Relationships with the Horse GIT Microbiome

Laminitis, EMS, and obesity are 3 main systemic disorders that can be interconnected. Laminitis is a metabolic disorder that may appear notably when huge amount of fructans (overgrazing) or starch from cereal-rich diets bypasses digestion in the small intestine and is transferred through the cecum to reach the colon. It causes great pain to horses and can lead to a failure of the attachment between the inner hoof wall and the distal phalanx of the foot, with dramatical consequences for sportive horses (including euthanasia). The exact physiopathology of laminitis remains unclear; however, it has been associated with previous complex microbial events leading to high blood levels of lactic acid, activin A, Lipopolysaccharide (LPS) and histamine [41]. EMS is an important and increasingly common clinical syndrome in horses and ponies. It has been defined by the American College of Veterinary Internal Medicine as “the presentation of a phenotype of obesity, insulin resistance and laminitis or a predisposition to laminitis in equine” [105]. Since this first definition, a recent paper proposed a revised definition stating that “EMS is not a disease *per se* but rather a collection of risk factors for endocrinopathic laminitis” [106]. The key and consistent feature related to EMS is insulin dysregulation, comprising fasting hyperinsulinemia, tissue insulin resistance and prolonged hyper-insulinemic response after a carbohydrate challenge. EMS is commonly associated with obesity and fat accumulation [106], which can in turn result in adipokine and cytokine dysregulation and to a pro-inflammatory status, contributing to apparition of insulin dysregulation. Obesity is not always synonymous of EMS, but obesity is clearly a risk factor, and is characterized by significant increase in blood cortisol and a correlation between blood leptin and body condition score [107]. Other blood biomarkers can be high triglycerides, free fatty acids, glucose and insulinemia. Although it is not easy to evaluate the prevalence of EMS, obesity prevalence is estimated at between 19 and 40% [108,109,110], while hyper-insulinemia is found in between 22 and 29% of equine populations [111].

Few studies have begun to explore relationships between those metabolic disorders and GIT microbiome in horses. The small number of studies and the use of different methodologies have prevented from a clear consensus (Table 3). Nevertheless, some key bacteria could play important roles and could serve as biomarkers. First, a decrease in alpha-diversity while a strong presence of lactobacilli (including *Lactobacillus delbrueckii*), streptococci and of *Megasphaera elsdenii*, a key lactate-using bacteria in the feces of horses suffering from Oligofructose (OF)-induced laminitis or natural EMS have been reported [41,112,113] concomitantly to a decrease in fecal pH (from −1.5 to −2.5 pH units post OF-Challenge) Interestingly, Elzinga et al. [113] reported a correlation between the greater abundance of *Megasphaera* and *Lactobacillus* and the blood 10E, 12Z-Octadecadienoic acid and asparagine that could be associated, as in humans, with disrupted epithelial barriers and metabolic alterations. *S. bovis*, *S. equinus*, but also different species of *Lactobacillus* (*Limosilactobacillus mucosae*, *Limosilactobacillus reuteri*, *Ligilactobacillus salivarius*, *Lactobacillus delbrueckii*, *Limosilactobacillus fermentum*) are key lactate producers and histidine decarboxylating bacteria [112,114]. Al Jassim et al. [115] found *L. salivarius* in stomach but also in the colon and rectal contents of OF induced-laminitis horses, suggesting that this bacterium, highly specific to the stomach of the horses, can grow all along the GIT and contribute to the high accumulation of lactic acid in the hindgut. Similarly, *L. delbrueckii*, described in humans as an important causative agent of D-lactic acid acidosis [116] have been found in rectal contents of horses. Interestingly, a bacterium sharing 97–98% sequence identity with *Mitsuokella jalaludinii* (Firmicutes, Veillonellaceae member) could also be an interesting biomarker as it has been found in the cecum and rectum of horses with OF induced-laminitis and is able to produce high rate of D-lactate. *Allisonella histaminiformans*, a very specialized histamine producing bacterial species, has been isolated from equine cecum and suspected to participate to the onset of laminitis [117] but was found only sporadically in the hindgut of horses with OF-induced laminitis [112]. Other potential biomarkers for those 2 metabolic disorders are *Akkermansia* sp., Ruminococcus/*Ruminococcaceae* as well as Veillonellaceae among which *Phascolarctobacterium* which decreased with laminitis and EMS ([41]; Table 3).

The studies made on obesity failed to find a consensus on association between obesity and alpha-diversity or Firmicutes/Bacteroidetes ratio, similarly to what has been reported in humans, making those parameters not valuable anymore. Indeed, alpha-diversity was unaffected [121,122] or even increased [18,107] in obese horses. Walshe et al. [123] reported an increase in alpha-diversity in obese horses subjected to a weight loss program. The abundance of *Lactobacillus* and the lactate concentrations decreased whereas the fecal pH increased in obese vs. lean horses, which is difficult to reconcile with EMS, although EMS-suffering horses are often obese. This underlines the complexity of those metabolic disorders. No consensus appeared on the relationships between *Lachnospiraceae*, *Ruminococcaceae* and obesity as their relative abundance increased or decreased depending on the study (Table 3). Those bacteria are specialized in the degradation of complex carbohydrates including fiber into SCFAs. Discrepancies about their pattern in obese patients also exist in rodents. Those families gather huge number of species and great metabolic versatility, and more research should be required to understand the behavior of those 2 key families, belonging to the common core microbiome of horses, in different situations, notably different diets. Remarkably, one OTU could be interesting as biomarker of obesity, namely *Butyrivibrio* sp., which correlated with different blood metabolites [107] and could play an important role in host-microbiome interactions. Finally, we would like to emphasize the importance of Fibrobacteres (*Fibrobacter* sp.) as key bacteria for all those metabolic diseases. Indeed, in almost all the studies using either obese, EMS or laminitis suffering horses, this phylum was reported to decrease. We thus strongly recommend understanding the biology of this bacteria in the equine gut ecosystem. *Rikenellaceae* family, which increased in obese and insulin dysregulated horses is also of interest. It is a saccharolytic family that has been shown to increase in obese mice [124]. Finally, Veruccomicrobia subdivision 5, now Kiritimatiellaeota, seems to play functional role although not consistent from one study to another and requires more attention. Overall, first studies allowed to highlight potential bacterial biomarkers. Of note, except *Fibrobacter* sp., those biomarkers are different from obese horses to horses suffering from laminitis or EMS, indicating that there are distinct microbial events between obesity and EMS/laminitis. The two latter are characterized by insulin dysregulation, which is probably a key event to study. Inflammatory process could be at the heart of all the physiopathology as insulin homeostasis is clearly related with inflammation status. In agreement with that, Zak et al. [125] found a strong correlation between blood basal insulinemia and serum concentrations of IL6. All in all, those results strongly suggest that targeting intestinal microbiome can be important for preventing equine laminitis, EMS and obesity.

## 5. Discussion: The Microbiota of Horses, a Crucial “Organ” Largely Associated with Health and Diseases in Horses

Our literature study confirmed that horses harbor a complex microbiota which evolves from the stomach to the colon and along the life of the animal. We also highlighted, as other previous authors, that the definition of a “compositional core microbiota” in horses is not easy and that a lot of research remains required as a large part of the bacteria that compose the microbiota is not yet well identified and described. Nevertheless, key facts are that the composition of the microbiota from the stomach to the colon corresponds to an oxygen and a pH gradient with more facultative anaerobes in the upper tract than in the hindgut. The upper tract of horses is indeed mainly inhabited by Proteobacteria and lactic acid bacteria, while the microbiota of the hindgut is specialized in the fibrolytic activity, with strictly anaerobic bacteria which are highly pH and oxygen sensitive. Thus, maintaining appropriate physico-chemical parameters to have a well-balanced gut microbiota all along the digestive tract is probably critical for the health of horses.

Gut health is a multidimensional concept related to diet, host and microbiota in which structure and functioning of the gastrointestinal barrier, gut microbial profile, and diet composition are continuously interacting (Figure 5). A stressful situation (e.g., sport, transportation), a high starch diet, or a digestive disorder may lead to an alteration of the gut environment, to higher gut permeability, to inflammation and to a change in the gut microbiota profile. Epithelial metabolism is supposed to play a crucial role as intersection between gut microbiome, immune cells and epithelial permeability and regeneration, as demonstrated in humans suffering from obesity, diabetes, or intestinal bowel disease [126,127,128]. The intestinal epithelium represents the frontline of the complex pathogenesis, lying at the interface of luminal inflammatory triggers such as the microbiome and host immune cells, and a breach of this well-structured barrier is suggested as cornerstone of chronic inflammation. Differentiated colonocyte is supposed to shape the microbiota, notably by performing mitochondrial β-oxidation to use fatty acids (mainly butyrate) as a source of energy and consumption of oxygen through oxidative phosphorylation; thus, creating a hypoxia in the intestinal lumen, which in turn will favor strict anaerobes growth, and fibrolytic activity to produce SCFAs, creating a virtuous cycle for the colon health. However, this cycle can be interrupted by epithelial injury or by an external factor (e.g., antibiotic) and colonocyte metabolism can switch towards anaerobic glycolysis, leading to lactate, nitric oxide (transformed into nitrate used as electron acceptor by Proteobacteria) and oxygen release in the lumen, which will favor the growth of facultative anaerobes like Proteobacteria [126]. Remarkably, certain opportunistic and potentially pathogenic bacteria can influence the colonocyte metabolism to expand their colonization. Therefore, to understand how the availability of respiratory electron acceptors becomes elevated during gut dysbiosis or inflammation could be a new area of research to develop novel preventative or therapeutic strategies in horses.

Our bibliography aimed at understanding if some bacteria of horse microbiota could be used as biomarkers of a dysbiotic state. As demonstrated in other studies, we confirmed that the bacterial community of horses suffering from intestinal diseases is considerably different from that of their clinically healthy counterparts. Globally, Shannon index in feces appears as a poor and inconsistent indicator. Meanwhile, a proliferation of Proteobacteria and lactic acid bacteria in the feces are often associated with digestive disorders or with a stressor applied to horses. As what is observed in humans [126], the most consistent and robust ecological pattern observed during gut dysbiosis seems to be an expansion of facultative anaerobic bacteria belonging to the phylum Proteobacteria in horses. Therefore, the shift of bacterial communities from obligate to facultative anaerobes in horses subjected to different stressors or affected by intestinal disorders spawned the hypothesis that dysbiosis might be a ‘dysanaerobiosis’ caused by increased oxygen availability in the large intestine. This reinforces the idea of investigating the evolution of the redox potential all along the gastrointestinal tract of horses subjected to different conditions.

The decrease in relative abundance of Fibrobacteres, *Methanobrevibacter*, *Ruminococcaceae* and *Akkermansia*, all strict anaerobic microorganisms quite sensitive to the physico-chemical environment is also noticed in several cases. Following the evolution of those populations could thus be interesting to detect potential digestive/systemic disorders. Steinberg and Regan [129] reported that the quantification of methyl coenzyme M reductase α-subunit (mcrA) gene by real-time qPCR successfully quantified different phylogenetic groups of methanogens. Our group is currently working on the development of a chip able to quantify the potential activity of different fibrolytic bacteria based on a selection of appropriate expressed CAZyme genes in ruminants [130]. Further investigations with such qPCR-based quantification of methanogenic or fibrolytic bacteria and diagnostics of horse physiological conditions, such as colic, enteritis, and other metabolic diseases, could verify if the abundance of those bacteria in feces can be used to indicate horse intestinal health.

Our work allowed underlining the need to develop knowledge on microbiota for each animal species. Indeed, while the beneficial effects of *Lactobacillus* and *Bifidobacterium* are well documented in humans [131], the observations gathered in the current review may suggest that the role of these specific microbes is not the same as in omnivorous animals. Another possibility is that *Lactobacillus* and *Bifidobacterium* can be useful in foals, as their growth mainly relies on milk digestion, and for specific effects in the upper tract of adult horses (e.g., for sportive horses fed with high level of starch); while being not beneficial for the hindgut. Harlow et al. [132] indicate that exogenous lactobacilli *L. reuteri* could mitigate the negative effect of cereal grain fermentations on the microbial community of the fecal microbiota by increasing the count of lactate producing bacteria, but also lactate using bacteria. In all cases, more specific research is necessary to characterize the equine microbiota of various digestive segments, along with the effects of specific bacteria on the health of those sites.

Among potential indicators, the Firmicutes/Bacteroidetes ratio has been scrutinized in different studies. In case of colic, this ratio resulted in inconsistent evolution, either increasing [89] or decreasing [90]. We reported a decrease of this ratio after a deworming drug application, while unclear results are reported for horses suffering from laminitis or obesity, making difficult to give a biological relevance for this indicator. Interestingly, this ratio has also been discussed in humans, with the healthiest enterotypes carrying an abundance of Bacteroidetes among their fecal microbiota. But like for horses, no clear consensus is established today about the biological significance and the relevance of this ratio [133]. According to these authors, discrepancies are likely to be explained by interpretative bias originating from methodological differences in sample processing and DNA sequence analysis, by a poor characterization of the recruited subjects and by a weak appraisal of lifestyle-associated factors affecting microbiota composition and/or diversity. Moreover, another obvious reason is that Bacteroidetes and Firmicutes encompass a huge number of bacterial species with a lot of functional roles which may be redundant or overlapping between the 2 phyla. Weese et al. [32] proposed that the Firmicutes/Proteobacteria ratio could be an indicator of a fecal microbiota from horse suffering from colic. However, the pattern of the relative abundance of Firmicutes can vary with the time of fecal collection, again making difficult to use it as a robust indicator.

A last ratio that could be of interest could be the ratio between Proteobacteria and Fibrobacteres, or between Proteobacteria and *Methanobrevibacter*. These 2 ratios could materialize the oxygen and pH status in the colon, are more finetuned than ratios with Firmicutes or Bacteroidetes, and thus could give precious indication on the physiological status of the hindgut. This hypothesis would predict that restoring epithelial hypoxia in the colon could be a therapeutic strategy to rebalance the gut microbiota. Of note, the use of live yeast has been reported as a promising strategy to increase fibre digestibility or to restore the gut eubiosis in ruminants [134] and in equines [135]. It is possible that the beneficial effects described are linked to the fact that one of the main modes of action of live yeast is to scavenge oxygen, thus restoring hypoxia in case of challenge. To validate this ratio and other useful microbial biomarkers, understanding the co-occurrence patterns among the different bacterial species (and notably the balance between strict and facultative anaerobes) in different context and GIT physico-chemical conditions, will be required to investigate their role in health and diseased conditions in horses and, ultimately, to develop novel biotherapeutics to reduce the incidence of different types of digestive or systemic disorders.

## 6. Conclusions and Future Directions: An Exciting Field of Research to Support Digestion, Performance and Health of Horses, but Still a Lot of Challenges

Improving horse health through modulation of the microbiome appears as a promising open avenue but also a strategy that is part of a comprehensive, holistic approach to ensure high sportive performance, good digestion and colon fermentation, and ultimately wellbeing. In this context, pre- and probiotics, but also to the use of new microbial-derived products such as non-viable bacteria or yeast or bacterial/fungal compounds that are defined as postbiotics [136,137] are interesting microbiome-focused strategies and are used for treatment and prevention of gastrointestinal diseases, or even of more systemic syndrome (Figure 6).

Multidisciplinary approaches will for sure be required to learn about how those -biotics may influence and interact with the horse. In vitro models as alternatives to in vivo studies are of increasing interest nowadays in all studies that focus on animal physiology, health, and intestinal microbiota. Such models represent a good pre-requisite before performing in vivo trials. For instance, in vitro batch systems have used equine fecal samples as inoculum to monitor short-term hindgut fermentations [138]. The parameters set up in the in vitro system are critical to ensure optimal conditions for the microbial populations [139]. The ANKOM Daisy II fermenter, used for quite a long time in ruminants, has been also used to measure feed digestibility by fecal microbiota from horses [140,141] and donkeys [140]. Whereas the published studies report that feces can be considered as suitable as microbial inoculum for in vitro digestibility studies, further research comparing in vitro to in vivo data with more forage types and methods is still needed to really establish in vitro digestibility as a viable technique for estimating digestibility in horses [142]. The development and performance of a three-stage fermentation model designed to simulate bacterial communities in the equine large intestine has been proposed [143]. This continuous fermenter model has been designed to replicate the physico-chemical parameters and the microbiota of the equine cecum, right ventral colon, and left ventral colon. According to the authors, the model was metabolically functional and was able to maintain a bacterial community close to that found in the large intestine of equines, although it was pointed out that an exact replication of the in vivo microbiota could not be achieved [6]. Thus, as it is currently proposed for other mammal species such as pigs [144] or humans [145], this model may be interesting to evaluate and compare diets, drugs, or various -biotics effects on microbiota.

Thus, state-in-the art in vitro, and in silico techniques will have to be designed to uncover the effects of pre-, pro- and other biotics on their targets, and metabolomic tools to identify in vivo key microbial members or their secreted molecules that either mediate benefits on the horses or can be used as gut health biomarkers. All those tools will offer unprecedented insights into the functionality of all the pre-, pro-, post-, parapro-biotics and will allow to design specific products, complex consortia or develop next generation probiotics. At the end, our main goals will be (i) to translate the benefits observed during research into real life outcomes and (ii) to answer the needs of the horse industry and support the development of targeted personalized solutions to improve equine performance, wellbeing, and health. To achieve that, of course, in vivo studies remain required to integrate all the factors and demonstrate the benefit in the horse.

## Figures and Tables

**Figure 1 microorganisms-10-02517-f001:**
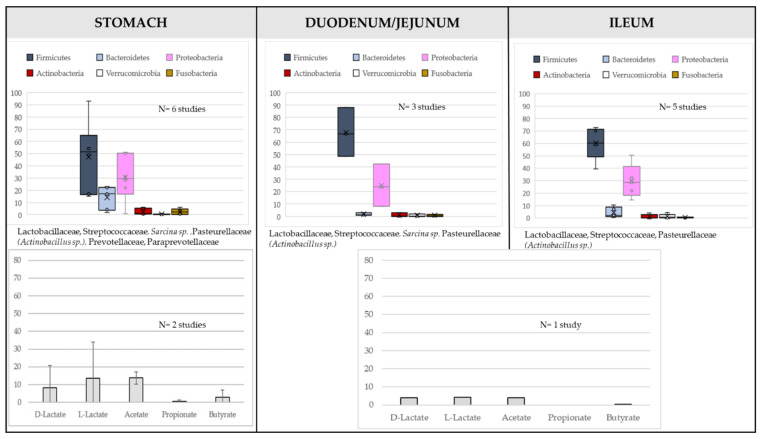
Main bacterial phyla and families (relative abundance, %) present in the upper tract of healthy adult horses and concentrations of the main organic acids found (expressed in mM).

**Figure 2 microorganisms-10-02517-f002:**
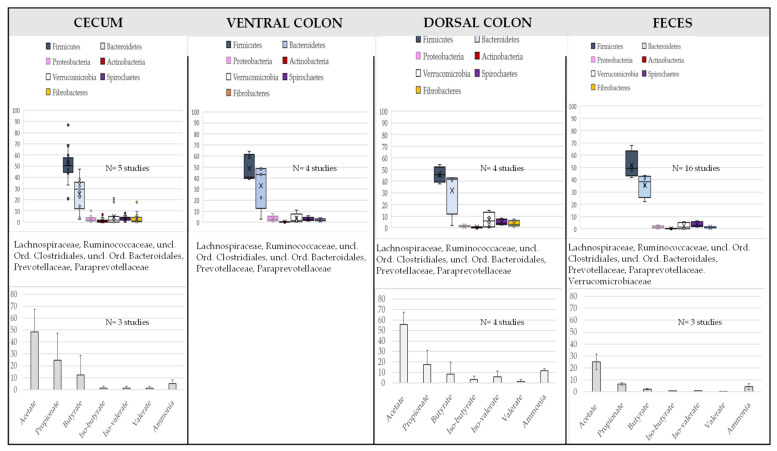
Main phyla and families (relative abundance, %) present in the lower tract of healthy adult horses and concentrations of the main organic acids found (expressed in mM).

**Figure 3 microorganisms-10-02517-f003:**
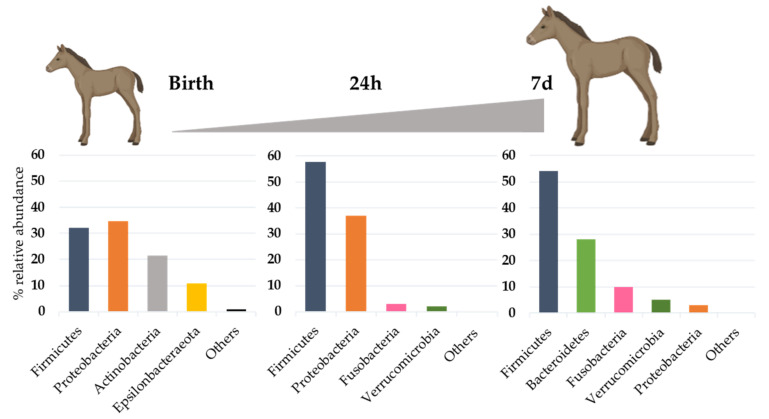
Evolution of the main bacterial phyla in foal rectal samples from birth to 7 days of age. After [51].

**Figure 4 microorganisms-10-02517-f004:**
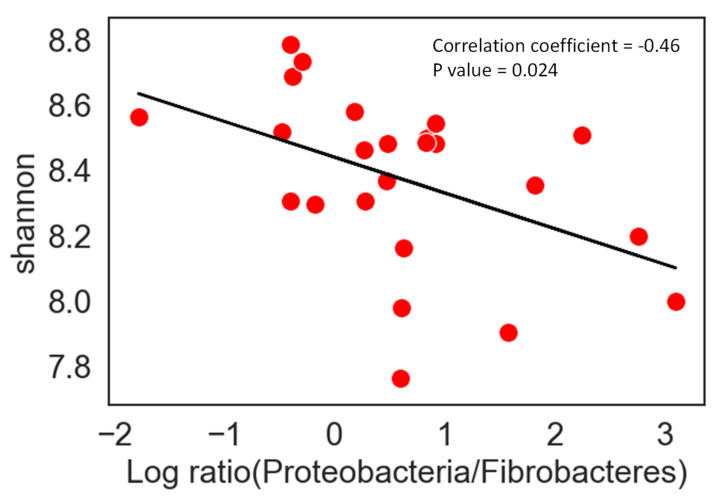
Correlation between the Shannon index and the Proteobacteria/Fibrobacteres ratio (expressed in log) in ponies and horses.

**Figure 5 microorganisms-10-02517-f005:**
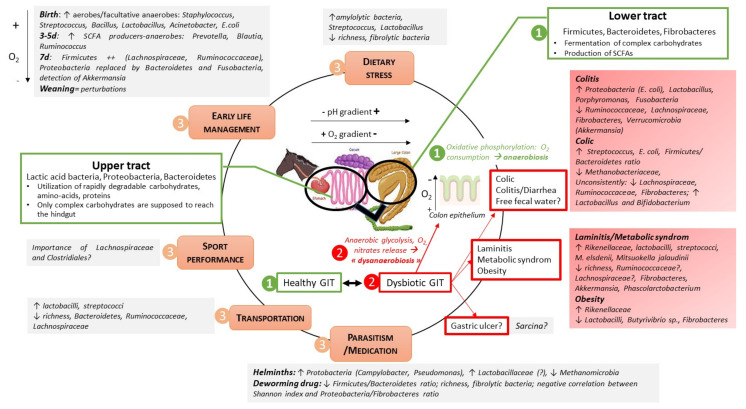
Factors contributing to GIT microbiota health or dysbiotic state, and identification of potential interesting microbial biomarkers. Green boxes: balanced microbiota (1); red boxes: dysbiotic states (2); orange boxes: factors that can alter the microbiota and lead to a dysbiotic state (3). Beige boxes gather the main bacteria that are affected by the dysbiotic situations. In this figure, the evolution of the oxygen gradient with age, GIT segment and at colonocyte level, as well as the importance of the crosstalk between colonocytes and microbes, and the possible switch in colonocyte metabolism are also indicated.

**Figure 6 microorganisms-10-02517-f006:**
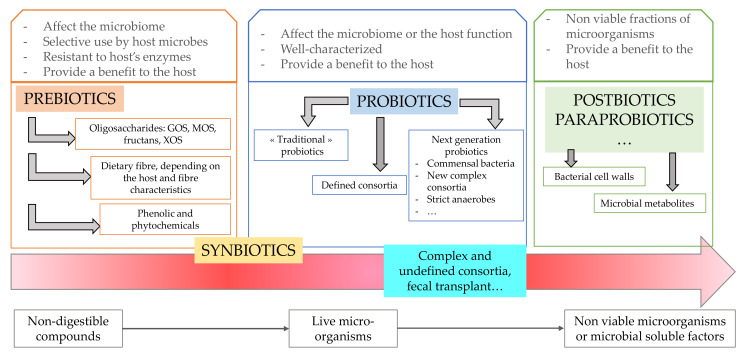
Possible strategies using biotics for horse nutrition and health applications.

**Table 1 microorganisms-10-02517-t001:** Main effects of high starch diet on hindgut microbiota.

Reference	Type of Diet and Protocol Details	Effect on Microbiota
[61]	4 fistulated horses in a 4 × 4 latin square design. Diets fed 2×/day. HF = 11.6% starch; 41% NDF HS = 30.1% starch; 30.7% NDF NDF:starch ratio of 3.5 for HF and 1.0 for HS diets Enumeration of cultivable viable functional bacterial groups in the cecum and colon	Increase in Lactobacilli in the cecum and the colon with HS diet but no change in total anaerobes, cellulolytics or streptococci
[60]	8 fistulated horses in a 4 × 4 latin square design. Diets fed 2×/day HS = 3.4 g/kg BW of starch per meal but maintaining NDF:starch ratio of 1.0 Enumeration of cultivable viable functional bacterial groups in the cecum and colon	With HS, total anaerobic and lactic acid-utilizing bacteria increased, and cellulolytic bacteria decreased in the cecum. Increase in lactobacilli and streptococci both in the cecal and colonic contents
[19]	17 mares Hay diet vs. hay plus a high cereal supplement (35% of starch in the high starch diet) 16S rDNA sequencing	With starch diet, increase in Proteobacteria phylum (*Succinivibrio*/Succinivibrionaceae related OTUs) Increase in *Phocaeicola* related OTU (Bacteroidetes phylum), increase in some *Lachnospiraceae* related OTUs but decrease in other *Lachnospiraceae* related OTUs (Firmicutes phylum)
[66]	6 fistulated geldings in a 2 × 2 latin square design. Diets fed HS = 56%/44% hay/barley diet for 3 weeks (0.20% BW of starch per meal) HF = 100% hay 16S rDNA sequencing Enumeration of cultivable viable functional bacterial groups in the cecum and colon	Reduced bacterial diversity with HS Impact of HS diet on community composition: decrease in *Ruminoclostridium* genus in the cecum, decrease in Bacteroidales S24-7 and, *Lachnospiraceae* NC2004 groups, increase in Veillonellaceae family in the colon Total anaerobes, starch utilizers, lactate utilisers increased, and cellulose utilizers decreased
[65]	Ten 18-month-old ponies in a 2 × 2 cross-over design with 2 experimental diets HF and HS. Diets fed 2×/day. HF = hay and lucerne; 0.46 g/kg of BW of starch per meal HS = hay and compound mix; 0.96 g/kg BW of starch per meal 16S rDNA sequencing in the feces	Bacterial diversity lower in HS diet with higher variance Impact of HS diet on community composition: decrease in *Ruminococcaceae* family abundance and increase in *Streptococcus* OTU
[18]	23 pony mares of different ages followed for 2 years HF: Hay diet at 2% body mass as daily dry matter intake for 4 weeks HS: 2 g starch per kg body mass distributed for maximum 5 days 16S rDNA sequencing in the feces	Diet transition increased *Candidatus*, *Saccharibacteria* and Firmicutes phyla abundance and reduced Fibrobacteres abundance At the genus level: *Streptococcus* abundance increased but not consistently across individual animals. Fecal pH and SCFA concentrations modified by diet but considerable inter-individual variation

**Table 2 microorganisms-10-02517-t002:** Main microbiota changes (alpha and beta diversity, taxonomic composition) measured in the GIT segments of horses according to different hindgut digestive disorders.

Reference	Horses	Microbiota Analysis	Alpha-Diversity	Beta-Diversity	Composition
**COLITIS and DIARRHEA**					
[34]	30 clinically healthy horses of two age groups (adult vs. geriatric) and 5 geriatric diarrheic horses	Fecal sample in rectum, V1-V2 region of the 16S rRNA gene	Not reported	Significant differences between healthy and diarrheic horses. Strong heterogeneity among the diarrheic horses	On average ↑ Proteobacteria and ↓ Fibrobacteres and *Verrucomicrobia*. Negative correlation between relative abundance of *Verrucomicrobia* and diarrhea score
[88]	Cecal and colonic tissues from 7 horses with acute diarrhea (*post-mortem*) of 3 horses free from digestive diseases (chronic arthritis and cervical stenosis) differing for sex, age, and breeds	V3-V4 region of the 16S rRNA	No difference colitis vs. control in both colon and caecum. In colitis horse, mucosa richness > content richness	Significant difference in mucosa and content in colon and caecum	Regardless of the intestinal compartment (colon or cecum) or the sampling site (luminal or mucosal), there were 27 taxa associated with healthy horses (LDA > 3) and 24 taxa associated with horses with colitis. ↑ *Lactobacillus*, *Escherichia*/*Shigella*, Enterobacteriaceae and *Fusobacterium* and ↓ *Fibrobacter*, *Lachnospiraceae* uncl., *Clostridiales* uncl., *Fretibacterium* and Bacteroidetes uncl.
[30]	6 horses with chronic or acute colitis and 2 healthy donors to evaluate fecal microbial transplantation	Fecal swab and feces. V4 region of the 16S rRNA	↓ richness between donor and diarrheic horses	No difference	*↑ Intestinimonas*, unclassified Lactobacillales, *Lactobacillus*, and *Streptococcus*, when compared to the donors ↓ relative abundance of the genus *Saccharofermentans*
[27]	10 healthy and 10 diarrheic horses differing for sex, age and breeds	Negative for *C. difficile* by feces culture. Rectal swabs for fecal collection, V1-V3 region of the 16S rDNA	↓ richness and evenness; ↓ Shannon index	Significant difference	*↑ Actinobacillus*, *Porphyromonas*, *Roseburia* ↓ RC9 gut group and *Ruminococcaceae* unclassified
[30]	6 healthy horses and 10 colitis horses differing for sex, age and breeds	Negative cultures for *Salmonella* spp, as well as single negative fecal ELISA results for *Clostridium perfringens* enterotoxin and *C. difficile* toxins A and B. Fecal samples. V3-V5 Region of the 16S rRNA Gene	No significant differences in alpha-diversity	Significant difference	↓ Actinobacteria and Spirochaetes; ↑ Fusobacteria among which *F. necrophorum* and *F. nucleatum*; ↓ Clostridia, Heliobacteriaceae, *Lachnospiraceae*, Eubacteriaceae, *Peptococcaceae*, *Clostridiaceae* and *Ruminococcaceae*. Among *Clostridiaceae*, ↓ *Trepidimicrobium* and *Clostridium*
**COLIC**					
[89]	28 horses showing signs of colic into two study groups: horses with large intestinal colic (LC, *n* = 20) and horses with small intestinal colic (SC, *n* = 8). 24 clinically healthy adult horses. All horses were thoroughbreds	Fecal samples at D0 (admission) Amplicon sequencing of the V3-V4 region	↓ number species observed and Shannon index in horses with large intestine colic	Significant difference between control, large colon and small intestine colic horses	**Horses with small intestine colic**: ↑ Firmicutes, ↓ Methanobacteriaceae and subdivision 5 *Verrucomicrobia*. LEfSe: ↑ *Enterococcus*, *Lactobacillus*, *Acinetobacter*, *Bifidobacterium*, *Kurthia*, *Weissella*, *Rummeliibacillus*, ↓ *Methanobrevibacter*, *Coprococcus*, *Faecalitalea*, *Treponema*, *Akkermansia* **Horses with large intestine colic**: ↑ Bacteroidetes, *Lachnospiraceae*, *Streptococcaceae*, *Lactobacillaceae* and *Coriobacteriaceae*, ↓ *Verrucomicrobia*. LEfSe: ↑ *Enterococcus*, *Acinetobacter*, *Lactobacillus*, *E.coli*/*Shigella*, *Blautia,* ↓ *Methanobrevibacter*, Unclassified bacteria *Verrucomicrobia*
[90]	17 horses; 3 horses young, 8 mature and 6 as geriatric. 10 Thoroughbreds, 6 Warmbloods and 1 mixed-breed horses. Fourteen horses were admitted with a colic episode < 60 h and three horses were admitted with a history of colic ≥ 60 h. Different lesions of intestine	Fecal samples at D0 (admission), D1 and D3. Amplicon sequencing	↓ number of species observed between D0 and D3; = D0 to D1; = D1 to D3 ↓ number of species observed and Shannon index in horses with colic ≥ 60 h	Significant difference depending on the time	↓ Firmicutes from D0 to D1 and remained lower than admission on D3 Bacteroidetes and Proteobacteria ↑ in all horses, while Fibrobacteres ↓ from D0 to D3 in horses with colic ≥ 60 h. More profound changes in all horses with colic ≥ 60 h.
[91]	9 horses with large intestinal forms of surgical colic and orthopaedic controls with general anaesthesia same initial antimicrobial and analgesic protocol than colic horses	Colonic and fecal samples at the admission D0, fecal every 2–3 days during hospitalization, weekly during the first month after hospital discharge and then every 2 weeks for a further 2 months. Amplicon sequencing V1-V2 regions	No significant differences in alpha-diversity of fecal microbiota between colic and control horses at admission	No significant differences in beta-diversity of fecal microbiota between colic and control horses at admission	↑ 21 OTUs (mainly Fibrobacteres (*n* = 8), Bacteroidetes (*n* = 5) and Spirochaetes (*n* = 6)) ↓ 25 OTUs (Firmicutes (*n* = 9) and Bacteroidetes (*n* = 16)) in the fecal microbiota of case horses
[32]	*Post-partum* colic: 13 mares that developed colic, 13 mares that did not display colic and 5 nonpregnant controls	Fecal samples were collected approximately 14 D prior to the estimated foaling date, within 4 D after parturition, and 14 and 28 D after foaling. Episodes of colic were recorded. Amplicon sequencing of the V4 region	No significant differences in alpha-diversity of fecal microbiota neither in richness nor in evenness	Difference from 10 D before colic appearance	**In the >10 D previous the colic:** ↓ Firmicutes, ↑ Proteobacteria (↑ *Rhodopseudomonas*, uncl. Enterobacteriaceae and *Enhydrobacter*); **<10 D before colic appearance:** ↓ Firmicutes (↓ Sphingobacteriales, *Acetovibrio*, *Ruminococcus*), Bacteroidetes (uncl Bacteroidales) and Tenericutes. Firmicutes:Protebacteria relevance ratio **Shorter time (<4 D before colic)**: ↓ *Ruminococcaceae* and *Lachnospiraceae*
**FREE FECAL WATER (FFW)**					
[92]	Case-control study with 100 healthy and 100 horses with FFW differing for sex, age and breeds	Fecal collection, 3 periods: Oct/Nov; Dec/Jan; Feb/March. Culture to determine the concentration of *C. perfringens* and *C. difficile* + V3-V4 regions of the 16S rRNA gene	No change in richness, evenness and in Shannon index	No change	Negative to *C. perfringens* and *C.difficile*. 14 genera differed in relative abundance between case and control horses within at least one sample collection. These genera belonged to the phylum Bacteroidetes (*n* = 2, ↑ *Alloprevotella* and 1 Bacteroidetes), Euryarchaeota (*n* = 1; ↓ *Methanobrevibacter*) and Firmicutes (*n* = 11; ↓ *Bacillus*, *Solibacillus*, *Lachnoclostridium sensu stricto*, *Roseburia*; ↑ *Lactobacillus*, *Marvinbryanttia*, *Oribacterium, Ruminococcaceae UGC005, Saccharofermentans*). Overall no big changes in fecal microbiota composition and diversity.
[93]	10 horses with FFW for >12 months vs. 10 healthy horses differing for sex, age and breeds	Rectal collection of feces. V4 region of the 16S rRNA	No change in richness, evenness and in Shannon index	No change	No change
[94]	16 horses with FFW; 15 healthy horses differing for sex, age and breeds	1 fecal sample in spring, another in autumn. V4 region of the 16S rRNA gene	No change in richness, evenness and in Shannon index	No change	Differences in microbial community composition based on time point and health status were not observed on any taxonomic level.

**Table 3 microorganisms-10-02517-t003:** Main microbiota changes (alpha and beta diversity, taxonomic composition) measured in the feces of horses suffering from laminitis, equine metabolic syndrome, and obesity. PICRUSt = functional prediction for the 16S rRNA marker gene sequences. Functional metagenomes for each sample were predicted from the Kyoto Encyclopedia of Genes and Genomes (KEGG) catalog and collapsed to a specified KEGG level. LEfSe (Linear discriminant analysis Effect Size) determines the features (organisms, clades, operational taxonomic units, genes, or functions) most likely to explain differences between classes by coupling standard tests for statistical significance with additional tests encoding biological consistency and effect relevance.

Reference	Animal Characteristics	Microbiome Measures	Main Results vs. Healthy Horses
**Laminitis**			
[33]	10 normal horses and 8 horses with chronic laminitis	Feces. Amplicon sequencing, V5-V9 regions	↑ number of OTUs and ↑ 2 OTUs of *Clostridiales* order Great inter-individual variability
[112]	5 horses with OF-induced laminitis	Culture and molecular methods	*↑* complex *Streptococcus bovis*/*equum,* then of *Lactobacillus* sp. *E. coli* increased post-laminitis.
[118]	20 horses, 8 control, 6 with acute laminitis induced by corn starch infusion, 6 with acute laminitis induced by OF infusion.	Feces. Amplicon sequencing, V4 region from cecum	No evaluation of alpha and beta-diversity ↑ Firmicutes, *Lactobacillus*, *Streptococcus*, *Veillonella*, *Serratia* ↓ Bacteroidetes, Bacteroidales, *Bacillus* and *Solibacillus, Verrucomicrobia*, *Akkermansia*, *Ruminococcaceae* and Veillonellaceae
[41]	10 healthy horses, 6.7 y, 3 males, 7 mares; OF-induced laminitis	Feces. Amplicon sequencing, V4 region from feces Metabolomics from intestinal contents	↓ fecal pH Alpha-diversity: ↓ Beta-diversity: ≠ ↓ Kiritimatiellaeota, Fibrobacteres, Tenericutes, Lentisphaerae, Elusimicrobiae, *Verrucomicrobia*, Planctomycetes ↑ *Lactobacillus*, *Megasphaera*, *Allisonella* *↓ Fibrobacter*, *Phascolarctobacterium*, *Papillibacter*, *Alloprevotella*, *Candidatus soleaferrea*, *Oribacterium*, *Akkermansia*, *Elusimicrobium* Biomarkers (Lefse): *Lactobacillus* (*L. gasseri* and *L. delbrueckii*)*, Megasphaera* (*M. elsdenii*), *Sharpea*, *Streptococcus*, *Prevotella*-sp_DJF_CP65 Metabolites: ≠ clusters, 53 and 83 metabolites with higher and lower concentrations respectively. Enrichment of ABC transporters, glycerophospholipid metabolism, inflammatory mediator of TRP channels, lysine degradation, vitamin digestion and absorption, tyrosine metabolism Correlation network: asparagine and 9-hydroxy-10E, 12Z-octadecadienoic acid correlated + with *Lactobacillus* and *Megasphaera*
**EMS and insulin dysregulation**			
[119]	16 mixed-breed ponies classified according to their insulin dysregulation (5 healthy (NID), 11 medium (MID) to severe insulin-dysregulated (SID)) subjected to a dietary change: adding pasture to a hay diet	Feces. Amplicon sequencing V3-V4 regions, feces	Alpha-diversity: = except evenness, lower in MID ponies than in NID and SID. Beta-diversity: = ↑ Firmicutes and Bacteroidetes in MID (with higher blood GLP-1 concentration) ↓ Christensenellaceae R-7 group, ↑ Rikenellaceae and Kiritimatiellae in MID vs. NID
[113]	20 horses, mixed breeds and genders, 10 EMS; 10 non-EMS horses based on insulin dysregulation estimated through OST, general/regional adiposity and a history/predisposition to laminitis. Natural EMS	Feces. Amplicon sequencing, V4 region PICRUSt functional inference	Alpha-diversity: = Beta-diversity: ≠ PICRUSt data: = 12 significant OTUs (LEfSe): ↑ *Verrucomicrobia* subdivision 5 (now Kiritimatiellaeota), *Cellulosilyticum*, *Elusimicrobium*, *Clostridium* cluster XI and *Lactobacillus*, in EMS group; and ↓ *Fibrobacter*, Uncl. *Lachnospiraceae*, *Anaerovorax*, Uncl. Rhodospiracellaceae, Uncl. Flavobacteriaceae, *Saccharofermentans*, *Ruminococcus*
**Obese/Weight loss management**			
[18]	35 Welsh-section A ponies mares, 11 aged, 12 control and 12 obese. Glucose-insulin tolerance, digestibility evaluated in each phenotypic group. Controlled feeding, hay ration	Feces. Amplicon sequencing, V1-V2 regions, SCFAs, pH Metabolome (FT-IR spectroscopy)	Same copy numbers of bacteria, fungi and protozoa ↑ Shannon and Simpson index Beta-diversity: ≠ ↑ Bacteroidetes, Firmicutes and Actinobacteria, ↓ *Fibrobacter,* ↑ *Pseudoflavonifractor* ↑ fecal pH SCFAs: =, no change in fecal metabolome
[107]	78 horses: 24 lean, 17 normal and 37 obese	Feces. Amplicon sequencing, V4-V5 regions. Network construction between microbial OTUs and blood analytes	↑ Shannon index, chao1, observed OTUs and phylogenetic diversity Phyla level: = but ↑ Firmicutes: Bacteroidetes ratio in obese. 24 OTUs with different relative abundance between normal and obese horses BCS Obese positively correlated with Actinobacteria, Firmicutes (*Ruminococcaceae* and *Lachnospiraceae*) and Bacteroidetes Network: obese BCS positively associated *Campylobacter sp., Collinsella* sp., Prevotellaceae, *Selenomonas* sp., *Blautia* sp., *Mogibacterium* sp., *Adlercreutzi* sp., Erysipelotrichaceae, Propinibacteriaceae, *Butyrivibrio* sp., *Ruminococcaceae, Sutterella* sp.
[120]	10 Shetland ponies and 10 warmblood horses subjected to a 2-y body weight gain program	Feces. Amplicon sequencing, V3-V4 regions, SCFAs, lactate	*Ponies*: ↓ number of OTUs ↑ Firmicutes and Actinobacteria ↓ lactate *Horses*: ↓ Fibrobacteres, ↑ Actinobacteria; ↓ *Ruminococcaceae*, ↑ *Lachnospiraceae* ↓ isobutyrate and lactate
[121]	12 mature obese horses and ponies, mixed breed. 2 restricted diets for 16 w. (traditional concentrate + hay vs. nutrient balancer (hay only))	Feces. Amplicon sequencing, V1-V2 regions, 2 time points: 10 and 16 w.	Alpha-diversity: = Absence of Fibrobacteres at the start of the study Weight loss: ↑ *Anaeroplasma*, several unclassified Firmicutes and Bacteroidetes, *Anaerophaga*, *Phocaeicola* and ↓ *Lactobacillus*, *Streptococcus*, *Butyrivibrio*, *Roseburia* and uncl. *Acidaminococcaceae*
[122]	20 obese and 20 normal horses matched by farm origin. BCS >= 7: obese, 3 < BCS < 7	Feces. Amplicon sequencing, V4 region. Fecal metabolome and serum lipidome	Alpha-diversity: = Beta-diversity: = 8 OTUs ≠ among which ↓ *Methanobrevibacter*, *Ruminococcus*, uncl. *Lachnospiraceae*, uncl. Bacteroidetes (2 OTUs) 57 metabolites ≠: 18 higher and 39 lower in obese horses. ↑ isocitrate, citrate, aconitate TCA cycle, ↓ vit E metabolism 146 lipids affected, 110 higher and 36 lower in obese horses: ↑ free fatty acids 14:0, 16:1, 18:1, 18:2, 18:3 and of cholesteryl esters, diacylglycerols, phosphatidylcholines
[123]	14 overweighed horses and ponies, 2 groups: control and fed with a restricted diet (2 vs. 1.4% BW DMI), 6 weeks	Feces. Amplicon sequencing, V3-V4 regions. Fecal metabolome	Weight loss program: Alpha-diversity: ↑ Beta-diversity: ≠ ↓ *Eubacteriaceae*, *Pseudomonaceae*, ↑ *Coprococcus* and class of *Clostridia* in the treated group. No change in metabolome. Network: positive correlation between *Rikenellaceae* and urocanic acid, *Ruminoccocaceae* with propionate, and *Ruminoccocaceae* and *Phascolarctobacterium* with urocanic acid.

## Data Availability

Not applicable.

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
