# Peer review of "Gastro-Intestinal Microbiota in Equines and Its Role in Health and Disease: The Black Box Opens"

_microorganisms, 2022, doi:10.3390/microorganisms10122517_

Round 1

Reviewer 1 Report

The review is very interesting. It contains a lot of information useful to specialists in the field of horse metagenome study. It will also be useful to veterinarians and equine therapy developers. A lot of information has been collected from a large number of articles, but there is no link to the very good review "The gut microbiome of horses: current research on equine enteral microbiota and future perspectives" 2019 (doi: 10.1186/s42523-019-0013-3). The authors should include this review. It is also necessary to change the design of the figs to be more understandable and aesthetic for the reader. It is necessary to correct all the pictures, except for 5. You can take the design of the pictures in the review as a basis: doi 10.1186/s42523-019-0013-3

Author Response

We thank the reviewer for his(her) positive report. The reference cited by the reviewer was part of our literature database but we forgot to include it in our submitted manuscript. It is now cited lines 47, 58 and 235.

We understand the comment related to figures and we tried to improve their readiness and aesthetic format. We hope that under the new presentations, the figures will be better designed.

Reviewer 2 Report

Please add references in lines 86-89.

In line 92, delete this sentence “which are mostly active in the distal 92

parts of the gut”.

Line 99 ones not one

Lines 200-203, delete this sentence “Verrucomicrobia subdivision 5 (now 200

classified as the separate phylum Kiritimatiellaeota) may also have important functional

role (even if not clear yet) and has been reported to be present with high relative abundance in some studies [3,9]”

In line 213, add “the” before strong.

In line 226 pay attention “to” not “on”

In line 396, …….only in some animals,  not in horse? I am confused.

Question 1 Line 557:  What's the difference between the colitis and healthy horses in beta-diversity.

In lines 617-625: I think the F/B ratio is not related to the colic. I suggest deleting these sentences.

In the text, author should pay attention to the supplementation of definite articles.

Question 2

From birth, do horses already have microbes in their bodies from their mothers? Please add the reply in text.

Suggestion

The conclusion part should be reduced its length and it contains overlapping part in the text. 

Author Response

We thank the reviewer for his(her) positive report. We made the changes required as we could, and we put in yellow in the text those changes in order to make reading easy.

Below are the answers to the reviewer’s comments:

Please add references for lines 86-89:    We added a reference as recommended.

In line 92, delete this sentence “ which are mostly active in the distal parts of the gut”:      Sentence was deleted.

Line 99: ones not one:   Modification has been made.

Lines 200-203, delete this sentence “Verucomicrobia…with high relative abundance in some studies [3,9]:         Sentence is removed.

Line 213, add “the” before strong : Done.

Line 226 pay attention “to” not “on”: correction has been made.

Line 396 …only in some animals, not in horses? I am confused: Our sentence was indeed confusing, here we were referring to some individuals in the cited study, so we were still referring to ponies. We replaced “animals” by “individuals” to improve clarity, see line 392 of the revised version of our manuscript.

Question 1 line 557: what’s the difference between colitis and healthy horses in beta-diversity? Difference in beta-diversity highlight dissimilarity in microbiota structure of colitis and healthy horses as reported in the cited references. The more detailed differences in taxonomic composition are described lines 556-575.

Lines 617-625, I think that the F/B ratio is not related to the colic. I suggest deleting these sentences. The F/B ratio is commonly used as a marker of health or disease in several animal species and also in humans. An alteration of this ratio has been measured in several studies but we fully agree with the reviewer that it is not specific to colic. We pinpoint the fact that it may not be sufficient to evaluate the disease status and we think it is an important comment for the readers to increase awareness of the need to use more specific biomarkers.

In the text, authors should pay attention to the supplementation of definite articles. We reviewed the manuscript to check the relevancy of using definite articles and we modified the text accordingly.

Question 2. From birth, do horses already have microbes in their bodies from their mothers. Please add the reply in text.

As reported lines 268-278 of the initial manuscript, it is still unclear whether the neonate foal harbors live microbiota before the birth event. This microbiota (or microbial components at least) might be transferred in utero through dendritic cells to the placenta. What is more documented is that microbiota is transferred from birth from the dam to the offspring. We realize that this might not have been clear enough and we propose to add a sentence line 255 of the revised version.

Suggestion: the conclusion part should be reduced its length and it contains overlapping part in the text. We propose a shortened conclusion according to the reviewer’s suggestion, please see lines  978-1042 of the revised version.

Reviewer 3 Report

Characteristics and effect factors of the horse GIT microbiota were reviewed in this manuscript. The contents are abundant. There are many  interesting  results. I give following comments: 

1. Only genus and species should be italic, family such as Lactobacillaceae, Streptococcaceae and Clostridiaceae needn't be italic, please check all.

2. "is" or "are"? and “has” or "have"? should be checked.

3. What's "core microbiota"? It should be given reference.

4.Please differ microbiota in feces, intestinal mucosa and intestinal content, they have different functions.

5. All SCFA should be SCFAs.

6. "2.4. Environmental conditions" is very wide.

7. Why only birth, 24h and 7d is showed in figure3? "1.2. Evolution of the horse microbiome with age" should differed from diet change?

8. Is there difference between male and female?

Author Response

First, we thank the reviewer for the time spent. We made the changes required as we could, and we put in yellow in the text those changes in order to make reading easy.

1.Only genus and species should be in italic, family such as Lactobacillaceae, Streptococcaceae, and Clostridiaceae needn’t be italic, please check all. The text has been checked accordingly.

2.“is” or “are”, “has” or “have” should be checked. This has been done.

3.What’s core microbiota? It should be given reference. We have already given a definition of the core microbiota in the first version: A “common core microbiota” i.e., a group of microbial taxa that are shared by all or most horses [3]. However, we also added the definition from Dougal et al (2017) in the revised version.

  1. Please differ microbiota in feces, intestinal mucosa and intestinal content, they have different functions. We do not see in which part(s) of the text the reviewer would like to see these precisions. In the first part, each paragraph is dedicated to a particular GIT segment; and the differentiation between mucosa and content for the hindgut has been made (see line 165).
  2. All SCFA should be SCFAs. This has been modified in the revised version.

6.“2.4. Environmental conditions” is very wide. We have changed the title by “Heat stress conditions”.

  1. Why only birth, 24h and 7 d is showed in Figure 3. Evolution of the horse microbiome age should differ from diet change? We chose to represent the microbiota from birth to 7 days in the figure because, as stated in the text, at 7 days, there is a shift between Proteobacteria and Bacteroidetes (see lines 302 and 303) and that from this age, the microbiota of the young is quite close, although not completely similar to the one of the mares. We also explained in the text that indeed, the diet is an important factor that may change the horse microbiome at the early age, with notably a paragraph dedicated to weaning from line 320.
  2. Is there difference between male and female? To the best of the author’s knowledge, no report has established a difference between male and female related to the early microbiota colonization process.

Round 2

Reviewer 2 Report

The authors well revised the manuscript, and thus is suitable for publication.

Reviewer 3 Report

I have no new comment.